# Planning and Scheduling for Cooperative Concurrent Agents with Different Qualifications in the Domain of Home Health Care Management

**Colja A. Becker and Ingo J. Timm**

Business Informatics I, Trier University

Behringstrasse 21 - Campus II

54296 Trier, Germany

## Abstract

Rising demand for home health care services combined with a shortage of professionals leads to increasing workload of existing employees. However, there is a performance limit so that it will no longer be possible to offer these services maintaining quality and economic viability without changing operational management. To cope with this situation, we propose a concept of task bundle splitting options among several cooperative agents with different qualifications. Further, we integrate this concept in a planning and scheduling algorithm for multiple concurrent agent actions which can increase efficiency and can improve use of limited resources in operational processes. For this purpose, possible concurrent actions of agents with different qualifications were evaluated combined with the scheduling process and compared to alternatives. As a first step, this contribution presents the concept as well as an algorithm generating an optimal solution and gives an insight into future work.

## Introduction

Many countries face the challenge of coping with increasing demand for care services. For example, in Germany the number of people in need of care will rise by around 32 percent by 2030, resulting in a shortage of care personnel (Rothgang et al. 2016). Besides stationary facilities and the support of relatives, home health care (HHC) is one possibility to receive care services. Here, caregivers are equipped with cars and drive to the patients' homes to render the required services.

To cope with an increasing demand in HHC, additional caregivers must be hired by service providers. However, the availability of (professional) caregivers on the labor market is very limited. Rising demand for HHC services combined with a shortage of professionals leads to the problem that the workload of existing employees increases. There is a performance limit so that it will no longer be possible to offer HHC services maintaining quality and economic viability without a change in operational management. Following this, managing existing human resources in HHC gains in relevance to enable efficient employment.

Inspired by *cooperative multiagent planning* as well as task decomposition in *hierarchical task network planning*,

we propose a concept of task bundle splitting options among several cooperative agents (caregivers) with different qualifications. Further, we integrate this concept in a planning and scheduling algorithm for multiple concurrent agent actions which can increase efficiency and can improve use of limited resources in operational processes in the domain of HHC.

## Literature Review

Since an increase in efficiency and an improvement in using limited resources can rise coordination effort, the usage of methods from the field of *multiagent systems* seems suitable. Moreover, knowledge and scheduling issues have a distributed structure among the participants in the domain of HHC. Here, operational management processes in terms of planning and scheduling can be supported by multiagent systems as well as decision support systems using agent technology (Becker, Lorig, and Timm 2019).

López-Santana et al. developed an MAS combined with a *mixed integer programming* model which takes cargivers' qualifications into account such that corresponding scheduling and routing is achieved (López-Santana, Espejo-Díaz, and Méndez-Giraldo 2016). The approach aimed at minimizing travel times of caregivers as well as delays in arrival times at customer locations. Similar, the approach by Xie and Wang focuses on minimizing service costs by creating an initial schedule using an optimization model and the schedule will be updated periodically during runtime (Xie and Wang 2017). The latter is based on communication between agents and a central re-scheduling. The approach by Marcon et al. uses a global optimizer to assign each caregiver to a set of customers with a corresponding route proposal, which can be adapted later by the caregiver (Marcon et al. 2017). Following this, a scheduling and routing solution is given. Here, each caregiver interacts with his own patients, so interchangeability is not possible, and there is no coordination between caregivers in order to reach a better joint solution. By changing the local decision-making mechanisms, different higher-level objectives can be pursued, e.g., minimizing waiting times. Remaining approaches which support planning and scheduling in operational HHC management surveyed by Becker et al. provide information management, standard scheduling solutions, frameworks, communication platforms, and basic coordination solutions.

Considering all approaches, the use of several different

qualifications in conjunction with joint processing of subsets of tasks for better deployment of limited resources has not been considered so far.

## Problem Description

In cooperation with an experienced HHC provider, we analyzed different processes related to the operational management of service provision. This also includes planning and scheduling in this domain. The following description of the problem is derived from the associated observations.

A service provider employs a set of caregivers $C$ and each of them has his or her own level of education. These qualification levels are given by the set $Q$. Every service from the set of all provided services $S$ is assigned to a qualification level, too. Further, each service $s \in S$ has a specific duration.

$$u : C \to Q \qquad w : S \to Q \qquad Q \subset \mathbb{N}$$

Following this, executing a service by a caregiver requires a qualification level at least equal (or greater) to the assigned qualification level. Further, an HHC provider has a set of patients $P$, which is called customers, who request services for certain times of a day. For this purpose, a day is divided into different time intervals $Y$ based on the set $T$ of all time points of a day.

$$Y = \{ (y_{start}^n, y_{end}^n) \mid y_{start}^n, y_{end}^n \in T \ \wedge \ y_{start}^n < y_{end}^n \\ \wedge \ y_{end}^{n-1} < y_{start}^n \ \wedge \ y_{end}^n < y_{start}^{n+1} \}$$

For example, a day might be divided into an early shift, a midday shift, and a late shift. A customer is able to request a service for one or more shifts per day and may also request different services in the same shift. All possible customer orders (service requests) are specified by the Cartesian product of services $S$ and time intervals $Y$. Thus, using the power set the function $r$ expresses the demand of a customer $p \in P$.

$$r : P \to \mathcal{P}(S \times Y)$$

One order $(s, y) \in r(p)$ of customer $p$ will be assigned to a caregiver and a certain time window by the function $a$ which corresponds to an operational management task in order to generate a schedule for the considered day. The caregiver assignment has to satisfy the required qualification level $q \in Q$ and the time window must be completely within the time interval given by the corresponding shift $y \in Y$.

$$a : r(p) \to T \times T \times C, \quad (s, y) \mapsto (t_{start}, t_{end}, c)$$
$$\text{where} \quad c \in C \ \wedge \ y = (y_{start}, y_{end}) \\ \wedge \ y_{start} \le t_{start} < t_{end} \le y_{end} \\ \wedge \ u(c) \ge w(s)$$

The full schedule $Z$ is often created manually at present by an operational manager for each day in advance using simple scheduling support software without specific scheduling algorithms. A schedule entry contains a customer $p$ as well as one of his orders $x = (s, y)$ and an assigned caregiver $c \in C$ who has to render the requested service in the time window starting at $t_{start}$ and ending at $t_{end}$ at the respective customer's location.

$$Z = \{ (p, x, j) \mid p \in P \ \wedge \ x \in r(p) \ \wedge \ j = a(x) \}$$

In operations, caregivers are often equipped with mobile devices and corresponding software for knowledge sharing and documentation tasks. Customer data and related order data as well as central schedule data is linked with the software. By using these devices, caregivers know where to go and what to do. Meanwhile, process times for internal documentation associated with service provision at each customer location are automatically recorded by the mobile devices.

The environment is represented by a directed graph $g = (V, E)$ where each customer $p \in P$ is assigned to a definite node $v \in V$ and each node is linked to each other node by a directed edge $e \in E$. Further, each edge is assigned to a value which describes the related travel time. For simplicity, all caregivers start at the HHC office which is also represented by a node of the graph.

$$k : P \to V \qquad f : E \to \mathbb{N}$$

All services for a customer $p \in P$ in a single day's time interval can be referred to as a requested task bundle for this customer. More precisely, a task bundle is a subset of the assigned subset by the function $r(p)$, such that all primitive tasks (services) for the customer $p$ in a specific time interval on a certain day will be performed directly in sequence at the customer's node.

In the following, as a first step the concept will focus on only one time interval without certain time windows for orders. The accomplishment of all task bundles in this time interval is the goal of the problem, i.e. one task bundle for each customer, while minimizing the overall processing time. Depending on the number of caregivers $C$, the task bundles can be performed concurrently.

## Concept

According to the classification by Torreno et al., the concept presented below belongs to the conceptual scheme "Planning *for* multiple agents" (Torreño et al. 2018), in which *one* agent plans and $n$ agents execute the plan. In Addition, scheduling is done by using the duration of the actions $S$ as well as the travel times given by the function $f(e)$ for edges $e \in E$. The number of execution agents corresponds to the cardinality of the set $C$ and the planning agent can be seen as a central planning unit at the HHC office. As mentioned before, caregivers are cooperative agents, which share the same goal, and act concurrently. Further, all actions are considered to be deterministic. An action is either an element of the set $S$, which requires a certain qualification level $q \in Q$ or just a move action to get from one node of the graph $g$ to another one. Obviously, moving between nodes does not require a certain qualification level unlike rendering a specific service at a customer's node.

### Task Bundle Splitting

Different qualifications are provided by the agents $C$ and assigning each task bundle to exactly one agent, i.e. assigning each customer to only one caregiver, can result in idle times of some agents with certain qualifications and overload of others depending on the actual conditions, e.g., order situation, travel distances, and distribution of qualifications. Caregivers with a high qualification level are considered as

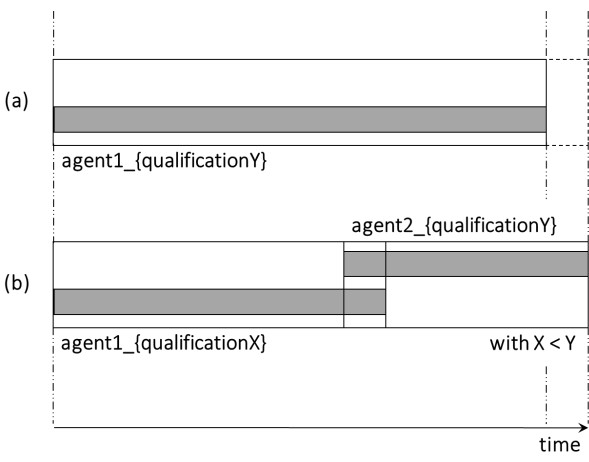

Figure 1: Two alternatives to accomplish a task bundle.

a very limited resource and it is assumed that more agents with a lower qualification level exist. Since some services in a task bundle can require a lower qualification level than other services in the same task bundle, assigning each task bundle to exactly one caregiver may require them to do some work for which they are overqualified. The latter can lead to error susceptibility and dissatisfaction among employees. Especially if caregivers with a high qualification level are highly requested due to the current order situation, the time for tasks of lower qualification levels is quite costly for these employees.

As a main part of this concept, we suggest to compare splitting of task bundles with the conventional procedure as an inherent component of the planning and scheduling process. In Figure 1, the two alternatives of processing a task bundle are depicted. The first variant (a) is the conventional procedure where one task bundle is assigned to exactly one agent. All contained tasks, consisting of moving to the corresponding node and rendering requested services, are performed one after the other by the assigned agent. Here, the agent's qualification level must be sufficiently high to perform all primitive tasks of this task bundle. The alternative (b) shows the splitting procedure. First, the lower qualified agent moves to the customer and makes the usual preparations as well as the other requested services according to its qualification. After completion of these tasks, the higher qualified agent arrives at the customer's location and the two employees exchange information about current customer-related content. The information exchange is encapsulated with a fix time value as a overlapping coordination task for each of these agents starting with the arrival of the second agent. Further, this joint task can be used for customer-related issues which require two caregivers at the same time, e.g., lifting the patient out of bed. In some cases, the second agent does not arrive seamlessly with the completion of the last task of the first agent, so the latter has to wait for the arrival of the second agent, and this time can be used for further concerns of the customer like different human needs or desires. After the joint coordination task,

the lower qualified agent moves on with its schedule and the agent with the higher qualification performs the high-qualification services. It is important to note that by accumulating all individual task durations in a task bundle, the entire splitted task bundle takes a greater duration value due to the additional coordination task. Since the agent with the higher qualification level is considered as a very limited resource, the splitted task bundle always starts with the lower qualified agent in order to avoid idle times of the more scarce resource. In addition, the whole customer service, i.e. the entire task bundle, should not be interrupted out of consideration for a humane treatment of the customer. Following this, leaving the customer's location before arriving of the second agent is not permitted to the lower qualified agent.

**Temporal Planning**

In this concept, planning and scheduling based on the order data for a chosen time interval is executed as *forward state space search*. The initial state contains order data and environment data. The order data comprises all task bundles for the selected time interval, i.e. one task bundle for one customer containing all primitive tasks $s \in S$. To include the customer's location, a node attribute is given for each individual task bundle. As mentioned before, in the goal state all task bundles are accomplished.

Since this phase of our research project neglects runtime complexity, the search for a goal state is performed as simple tree-based *breadth-first search* in which the scheduling process is integrated. In Algorithm 1, the pseudocode for planning and scheduling in the domain of HHC is presented. In order to process every state, a queue is used with a loop and generated successors are added to the queue. If a state still contains task bundles to be done, all possible actions of idle agents were gathered in according to each agent's qualification level. Here, an action means taking a task bundle which is not in progress and not accomplished so far. Further, every combination of the possible actions of all idle agents are computed. In this procedure, a combination contains only actions which are not already assigned to another agent in the same combination. Because agents are acting concurrently, we do not care about the order in this combination, so the term *combination* is used instead of *permutation*. Each generated combination represents a successor link and is then used to create further states. So, a state will be linked to a successor if the state is not a goal state and it contains one or more idle agents which can choose an action to perform. Otherwise, there is still work in progress, so any task bundle already have been done or is currently in progress and the successor state will be a goal state.

As mentioned before, in this concept scheduling is an inherent process using action durations. While creating a successor state by applying an action combination of the respective link, agents which are currently performing an action remain in their statuses and idle agents which are affected by the generated combination will be assigned to further work. Following this, a search procedure is conducted in order to find the next earliest event to set a time value for the state's clock. Such an event is either an accomplished task bundle of an operating agent, i.e. changing the status of an agent,

**Input:** initial state

**Output:** schedule/schedules $Z$

1  initialState ← init();

2  queue.add(initialState);

3  **WHILE** *queue.containsElement()* {

4      state ← queue.get(0);

5      **IF** *state.orderData.containsElement()* {

6          idleAgents ← getAgents($C$, state);

7          agentOptions ← ∅;

8          **FOREACH** $c \in idleAgents$ {

9              acts ← computeOptions(c, state);

10             agentOptions.add( (c, acts) );

11         }

12         sLinks ← combinatorics(agentOptions);

13         **FOREACH** $k \in sLinks$ {

14             successor ← createState(k);

15             successor.clock ← searchEvent();

16             queue.add(successor);

17         }

18     **ELSE**

19         goalStates.add(state)

20     }

21     queue.remove(state);

22 }

23 rStates ← minProcessingTime(goalStates);

24 minZ ← rStates.getSchedules();

25 **RETURN** *minZ*

**Algorithm 1:** Pseudocode for planning and scheduling of cooperative agents with concurrent actions.

or a buffer event which is introduced to allow for idle agents to perform an action as a second part of a splitted task bundle which require travel time by moving from one node to the other. By this means, all possibilities of idle agents to take action in time were covered. If one or more agents accomplish a task bundle, they will be added to the set of idle agents which is the starting point of the new state. There is a possibility that these agents may take further actions, and again, every combination of possible actions of all idle agents are computed and further successors are generated.

The possibility of splitting a task bundle is integrated as a part of an action combination, too. If respective qualifications among agents are available, generating action combinations comprises performing a task bundle conventionally by one agent as well as splitting a task bundle into two parts as described in the previous section. While splitting a task bundle, the first part of the lower qualified agent is directly integrated in the action combinations and the second part of this task bundle is added to the order data to allow for successor states to generate further action combinations at the corresponding time value.

In the end, all goal states' processing times are examined. As a result, one or more goal states containing an equal value as the minimum time value of all goal states are used to extract related schedule data. This output provides the optimal solution to the problem.

## Evaluation

The previously described concept has been implemented in the *Java* programming language. This allows for evaluation and further experiments. As a first step, a small example scenario was created and applied to the prototype. In the following, the scenario is presented together with the experiment results.

The example scenario comprises four customers and two caregivers. One agent has the qualification level "1" and the other agent has the qualification level "3". In Table 1, all services for this example are listed. This data fragment was extracted from real-world data. The second column shows the corresponding durations in minutes and the third column shows the required qualification levels. In Table 2, the example order data is given. Each line shows one task

Table 1: Service data used in the example scenario.

| S.-ID | Min. | Q. | Description |
|-------|------|-----|-------------|
| 1 | 3 | 3 | *Eye rinsing* |
| 2 | 5 | 3 | *Respiratory toilet* |
| 3 | 3 | 2 | *Glucose measurement* |
| 4 | 2 | 2 | *Injection of medication* |
| 5 | 8 | 1 | *Assistance with movements* |
| 6 | 5 | 1 | *Assistance with excretions* |
| 7 | 5 | 1 | *Assistance with bedding* |
| 8 | 25 | 1 | *Washing and dressing extended* |
| 9 | 17 | 1 | *Washing and dressing basic* |

Table 2: Example order data.

| Order-ID | Node | Services | Involved Q. Level(s) |
|---|---|---|---|
| 1 | 4 | { 5, 6, 9 } | 1 |
| 2 | 2 | { 1, 8 } | 1, 3 |
| 3 | 1 | { 4, 6, 7 } | 1, 2 |
| 4 | 3 | { 2, 3 } | 2, 3 |

Table 3: Ten best solutions of the experiment.

| Plan-ID | Rank | Processing Time | Splitting |
|---|---|---|---|
| 1 | 1 | 54 | 1 |
| 2 | 1 | 54 | 1 |
| 3 | 1 | 54 | 1 |
| 4 | 2 | 63 | 0 |
| 5 | 2 | 63 | 0 |
| 6 | 2 | 63 | 0 |
| 7 | 2 | 63 | 0 |
| 8 | 3 | 65 | 2 |
| 9 | 3 | 65 | 2 |
| 10 | 4 | 71 | 1 |

Table 4: Schedules of solution plan 3, 4, and 9.

| Plan-ID | Agent-ID | Time | Order-ID |
|---|---|---|---|
| 3 | 2 | 00 - 13 | 4 |
| 3 | 1 | 00 - 15 | 3_Part-1 |
| 3 | 2 | 13 - 21 | 3_Part-2 |
| 3 | 1 | 19 - 54 | 1 |
| 3 | 2 | 21 - 54 | 2 |
| 4 | 2 | 00 - 33 | 2 |
| 4 | 1 | 00 - 35 | 1 |
| 4 | 2 | 33 - 50 | 3 |
| 4 | 2 | 50 - 63 | 4 |
| 9 | 2 | 00 - 35 | 1 |
| 9 | 1 | 00 - 30 | 2_Part-1 |
| 9 | 2 | 35 - 44 | 2_Part-2 |
| 9 | 2 | 44 - 57 | 4 |
| 9 | 1 | 41 - 56 | 3_Part-1 |
| 9 | 2 | 57 - 65 | 3_Part-2 |

bundle requested by a customer at a certain node on the graph $g$, which represents the environment. For simplification, the HHC office as starting location for all agents was set to node 4. Further, all edges $e \in E$ were assigned to the value "5". So, moving from one node to another node takes 5 minutes. Because some of the entries of Table 2 contains services with different qualification levels, these task bundles can be splitted in the planning process. The last column shows the involved qualification levels for each task bundle to clarify the relationships. The time value for the coordination task of a splitted task bundle was set to the constant value "1" as a simple example.

The application of the prototype generates 30 goal states. In Table 3, some results of the experiment are given. Each line shows a goal state with its related processing time in minutes in decreasing order. So, the ten best solutions are shown in the table and the first three entries contains the shortest processing time. Further, the information about using task bundle splitting in a solution plan is given by the last column. If one or more task bundles are splitted in a plan, the line shows the number of splitted task bundles in this column otherwise zero, which corresponds to a conventional solution method without splitting. In order to give more insight into the comparison, the schedules of the solutions 3, 4, and 9 are given in Table 4. Note that the schedule entries' time intervals include times for moving from one node to another node at the beginning of each interval. For example, the first entry of the schedule for solution plan 3 contains the processing of task bundle 4 while driving to the related node takes 5 minutes and ren-

dering all services of this task bundle takes 8 minutes, which adds up to 13. During agent 2 accomplishes task bundle 4, agent 1 processes the frist part of the splitted task bundle 3, which continues until minute 15. After that, agent 1 has to wait three minutes until agent 2 arrives. This additional time can be used for unscheduled customer desires. When the second agent arrives, the joint coordination task takes one minute. Then, with the beginning of minute 19 the first agent moves on to the next node according to its schedule, while the second agent processes the second part of the splitted task bundle.

As shown in the result in Table 3, in this scenario an improvement of processing time in the amount of 14.3 percent can be achieved by using task bundle splitting (line 1-3) instead of the conventional solution method (line 4-7). The simple planning and scheduling algorithm works well for small scenarios, but takes too long for greater real-word scenarios. Nevertheless, the concept of task bundle splitting can be successful as shown above, so handling with greather real-world scenarios will be part of further work.

## Future Work

As a next step, we will work on reducing search space as well as using technologies for increasing performance. The former will focus on applying heuristics to the concept. The latter will focus on methods using GPU computational power. Moreover, we will investigate how more general existing planning techniques can deal with this problem. For further evaluation, we have already gathered real-world data in oder to examine further steps of our concept with order data, travel times, and more service data taken from the real-world domain of HHC. In addition, comparing our prototype to state of the art temporal planners will be part of further evaluation steps. Moreover, we are working on integrat-

ing a standard *planning domain definition language* (PDDL) into our prototype as well as extending the concept regarding planning and scheduling for several time intervals and with respect to different time windows of customer orders.

One of the biggest next steps in the long term will be the extension of our concept to a dynamic runtime solution. In current operational management in the domain of HHC, delays in operational processes result in overtime hours of employees and potential time gains in these processes cannot be used to compensate for time delays with other employees. In addition, caregiver outages and unplanned urgent customer requests are possible in daily operations and make efforts for efficiency more difficult. Hence, low cost flexible adjustment of individual tasks or schedules for adaptively dealing with a dynamic environment is desirable. Especially multiagent technology is known for offering flexible solutions and adaptive IT systems (Kirn 2006). Moreover, knowledge and scheduling issues have a distributed structure among the participants and taking up-to-date local data of the real world into account can be necessary to achieve a proper planning result.

## Dynamic Planning and Scheduling

To increase flexibility in caregivers' operations and efficiency in the use of resources, we further propose an agent-oriented framework for dynamic planning and scheduling, which will be described in the following. In Figure 2, the framework is depicted. Before the beginning of the day, initial planning and scheduling as presented in the previous sections provide the schedule $Z$. The connected database includes the current schedule and all information described before, e.g., customer orders for several time intervals. After computing an initial solution, this schedule can be modified by a dynamic planning and scheduling procedure. Especially during the service delivery process, the schedule $Z$ will be modified to cope with a dynamic environment. For this purpose, the database provides required information during runtime as well, e.g., assigned qualification levels.

The inner HHC system components and their environment can be distinguished into real-world and virtual layer. Each real-world caregiver $c \in C$ is represented by a software agent in the virtual layer and is able to communicate with other agents. Using caregivers' mobile devices, a distributed structure can be established. During the service delivery process, each caregiver agent reacts on environmental-based planning disturbances like delays in service executions or travel times. If the further compliance with the own schedule segment is at risk, the agent tries to modify its schedule by searching alternative plans on its own as well as in combination with coordination and communication with other agents. Alternatively, a central re-planning is initiated. In addition, during the service delivery process, an agent checks several group-related task lists of new urgent customer orders and computes possible schedule modifications to include one or more new requests like every other agent does. The schedule modification with the lowest costs for the entire group of caregivers will be chosen. Also positive schedule deviations are used to reach a better joint solution. For example, there is a greater saving of time while render-

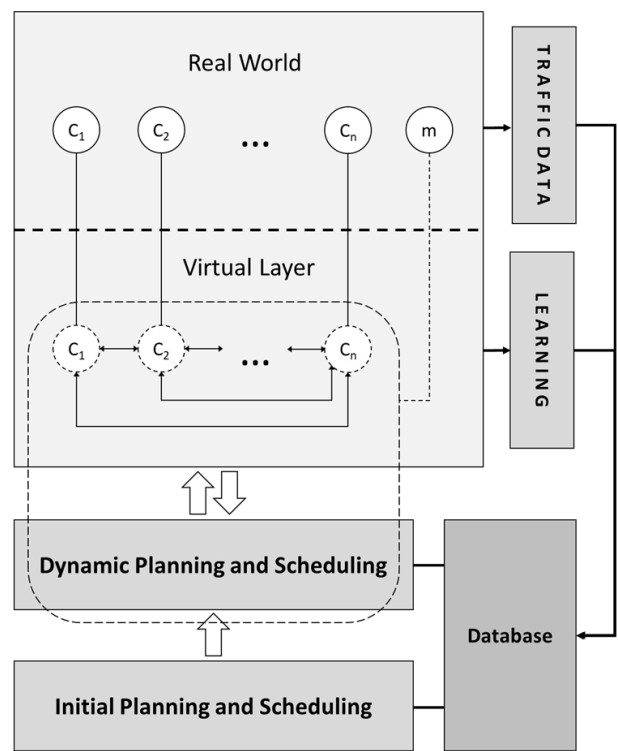

Figure 2: Agent-oriented Framework for Automated Dynamic Planning and Scheduling in HHC Management.

ing services at a customer's location, so the caregiver agent searches and compares alternative schedules under the new circumstances.

Caregivers of the real-world layer are continuously instructed with the next task bundle of the current schedule by their virtual agents using mobile devices. So, if something is changed in the background regarding scheduled tasks after the next task bundle, the caregiver does not have to worry about it, but simply continues to follow the instructions from one task bundle to the next.

Further, customers $P$, a road network including traffic, and the operational manager are parts of the environment of the real world. In Figure 2, the latter is referred to as $m$ and is capable of influencing the coordination between the caregivers. During the service delivery process, the manager filters new short-term customer requests and adds urgent requests to the group-related task lists mentioned before. Usually, customers with urgent medical issues call the HHC provider's office and the operational manager decides what to do. For every shift $y \in Y$ of the current day, a group-related task list containing new urgent customer orders exists and the lists are checked by the caregiver agents in order to assign new entries during runtime.

Furthermore, caregiver outages during the service delivery process are possible, e.g., car accidents or private emergencies of employees, but the medical care of customers have to be ensured. To this end, an affected caregiver can

use his or her mobile device to announce the outage and the virtual representative handles the allocation of his or her customer orders to the remaining caregivers. If the outage is announced to the office before starting the service delivery process the operational manager will just invoke the initial scheduling algorithm again.

## Using Advanced Data and Learning Mechanisms

Besides the data described previously, **further data** is necessary for planning and scheduling in order to generate better results in the long run. Initially, each service $s \in S$ is assigned to a time value which is required for basic scheduling issues. Furthermore, constraints based on different relationships between customers and caregivers exists. For instance, a female customer only wants to be treated by a female caregiver or a caregiver does not want to treat a specific person. Maintaining a long-term assignment of a caregiver to a customer instead of having alternating caregivers might also be in customer's interest which could increase scheduling effort. In addition, some caregivers do not want to perform certain services even though they have the appropriate qualification level. The reasons for this may vary, such as uncertainty due to lack of experience or physical aptitude. Beyond that, there are legal requirements for various aspects like break time specifications which are available in the database and must be taken into account in the scheduling process.

Regarding spatial aspects, the HHC office and all customer locations form a structure of nodes and weighted edges, which was introduced as the graph $g$. In this sense, static travel time matrices for different hours of a day are also stored in the database and they will be used for initial scheduling. During service delivery process, the traffic data module as shown in Figure 2 requests public traffic data for each edge using different real-world sources and updates edge weights at short time periods. Also the static travel time matrices will be updated periodically by the traffic data module. Traffic data and corresponding route data will be queried by caregiver agents during runtime for application in searching scheduling alternatives and in order to keep to the current schedule.

At runtime, different **learning mechanisms** working on the virtual layer generating additional data and update existing values in the database. Close to the stored traffic data, deviations related to certain routes are learned from caregiver agents' movement in the real world. For example, *reinforcement learning* can be applied to allow for a better routing in terms of cargiver's movement from one node to another using travel times for feedback information to the respective caregiver agent. Further, a value for deviations in service execution at customer's location is learned for each customer using automated documentation data from caregivers' mobile devices. Because recorded documentation times refer to entire customer visits instead of single service executions, a learned value is assigned to a set of services (task bundle). It is not uncommon for certain task bundles to be repetitively requested by a customer on a daily or weekly basis. The learned values can be used for other scheduling processes to obtain better planning results over time. With the approval of employees, these planning values can also be extended to include caregivers performing service execution. As a result, more differentiated values are available for planning and scheduling for each agent.

## Conclusion and Outlook

Increasing demand in the domain of home health care as well as a shortage of professionals faces the operational management with challenges regarding usage of limited resources and increasing efficiency while taking human needs and desires into account. For this reason, improvements of planning and scheduling issues in the domain of HHC are desirable. As a first step, we introduced a concept of splitting task bundles in temporal planning for cooperative agents with different qualifications. By applying this concept, concurrent processing of several task bundles can be improved due to better usage of limited resources as shown in an example scenario.

Future work will focus on further investigating what can *AI Planning* do for the described problem. Using this knowledge, we will extend our concept and reduce search space by applying heuristics. Furthermore, using real-world data and comparing our prototype to state of the art temporal planners will be part of next evaluation steps. In the long term, we aim at developing a dynamic planning and scheduling approach including the presented concept in order to increase efficiency in operational processes.

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
