# OpenReview forum: "Planning and Scheduling for Cooperative Concurrent Agents with Different Qualifications in the Domain of Home Health Care Management"
_icaps-conference.org/ICAPS/2019/Workshop/SPARK — SPARK 2019_

### Official Review · AnonReviewer2 · 2019-04-25
**An interesting Area**

**Rating:** 3
**Confidence:** 2

**Review:**

This paper focuses on the use of P&S techniques for Home Health Care Management. In particular, it addresses the problem of scheduling the work of a number of caregivers in order to deal with the current services demand. In a nutshell, the idea is to take a multi-agent perspective: each caregiver is a different agent, described by her own characteristics, that can cooperate with others in order to satisfy the overall demand. The core of the method relies on the possibility to break tasks between 2 agents with different qualification levels.
Some ideas to address the dynamic aspects of the problem at hand are also introduced and discussed.

The paper is generally easy to follow: the addressed problem is formalised, and an ad-hoc algorithm is proposed to deal with it. On this matter, I'd have preferred to get more information about the P&S techniques, and possibly the use of more general (domain-independent) approaches. As it stands, it does not give too much to the SPARK community, as I struggle to see how the proposed approaches can be used in different domains. I'm wondering, for instance, how good would a SLS approach perform.

All in all, the paper deals with an interesting problem, and can be of interest for the SPARK community. For future work in the area, I'd suggest the authors to either (a) focus on SLS or MIP approaches, or (b) investigate how more general existing planning techniques can deal with the specific problem. In particular, (b) would potentially provide insights and lessons for the wider ICAPS community.

---

### Official Review · AnonReviewer1 · 2019-05-01
**Work in progress**

**Rating:** 2
**Confidence:** 2

**Review:**

Home Health Care is growing which in near future would increase demands for health care workers and their efficient management. The paper proposes a method for scheduling health care workers while considering their levels of qualification.

Although the work is well motivated, it is still in a preliminary stage. The proposed algorithm seems to be quite naive as it explores all possibilities of task allocation to workers. It is not clear how long it takes (CPU-time) to generate plans/schedules by the proposed method. Only "Processing Time", which refers to plan/schedule make-span, is reported. Hence, at this stage it is not clear how the method can be leveraged in practice.

A large portion of the paper is devoted to future work. Using PDDL for modelling the problem might not be straightforward as time windows, modelled by numeric fluents, are not much supported by existing planners. A feasible alternative might be to consider classical planning such that time is discretized into time-stamps that can be represented as objects. Coping with dynamic environment (e.g. new care request, delays) is indeed an important aspect to consider (in future work). The part involving machine learning is not very clear. For example, if a customer want a specific care worker, then such a request can be encoded into the model. The only aspect which, in my opinion, can be well addressed by machine learning is predicting and dealing with unexpected changes in the environment (e.g. delays).

In summary, the work reported in the paper is still very preliminary, although I believe it has a potential.

---

### Decision · Program_Chairs · 2019-05-09
**Acceptance Decision**

**Decision:**

Accept

**Comment:**

Reviewers are agreed that this is an interesting problem for P&S technologies, and that the paper is well-written. The authors should pay attention however to the reviewers' request for more details and applicability of P&S techniques in the final version.